# Ultralong π-Conjugated Bis(terpyridine)metal Polymer Wires Covalently Bound to a Carbon Electrode: Fast Redox Conduction and Redox Diode Characteristics

**DOI:** 10.3390/molecules26144267

**Published:** 2021-07-14

**Authors:** Kuo-Hui Wu, Ryota Sakamoto, Hiroaki Maeda, Eunice Jia Han Phua, Hiroshi Nishihara

**Affiliations:** 1Department of Chemistry, Graduate School of Science, The University of Tokyo, 7-3-1, Hongo, Bunkyo-ku, Tokyo 113-0033, Japan; r.s@scl.kyoto-u.ac.jp (R.S.); h-maeda@rs.tus.ac.jp (H.M.); eunicephua@hcis.edu.sg (E.J.H.P.); 2Department of Chemistry, National Central University, 300 Jung-Da Rd. Jhong-Li 32001, Taiwan; 3Department of Energy and Hydrocarbon Chemistry, Graduate School of Engineering, Kyoto University, Nishikyo-ku, Kyoto 615-8510, Japan; 4Research Center for Science and Technology, Tokyo University of Science, Chiba 278-8510, Japan

**Keywords:** electropolymerization, coordination polymer, electron transfer, modified electrode, redox, hetero-metal complex, diode

## Abstract

We developed an efficient and convenient electrochemical method to synthesize π-conjugated redox metal-complex linear polymer wires composed of azobenzene-bridged bis(terpyridine)metal (**2-M**, M = Fe, Ru) units covalently immobilized on glassy carbon (GC). Polymerization proceeds by electrochemical oxidation of bis(4′-(4-anilino)-2,2′:6′,2″-terpyridine)metal (**1-M**) in a water–acetonitrile–HClO_4_ solution, affording ultralong wires up to 7400 mers (corresponding to ca. 15 μm). Both **2-Fe** and **2-Ru** undergo reversible redox reactions, and their redox behaviors indicate remarkably fast redox conduction. Anisotropic hetero-metal-complex polymer wires with Fe and Ru centers are constructed via stepwise electropolymerization. The cyclic voltammograms of two hetero-metal-complex polymer wires, GC/[**2-Fe**]–[**2-Ru**] (**3**) and GC/[**2-Ru**]–[**2-Fe**] (**4**), show irreversible redox reactions with opposite electron transfer characteristics, indicating redox diodelike behavior. In short, the present electrochemical method is useful to synthesize polymer wire arrays and to integrate functional molecules on carbon.

## 1. Introduction

Molecular wires are long molecules that allow electrons or holes to flow smoothly within them, thereby acting as molecular-scale, electrically conductive wires. They are one of the most important components of molecular-scale electronics for molecular or quantum computing [1,2,3,4,5,6,7,8,9]. However, their preparation faces two major challenges: achieving precise control of their lengths and structures when immobilized on electrodes and synthesizing wires long enough to link and integrate large numbers of functional molecular units to obtain high-performance devices [10,11,12,13,14,15,16,17,18,19,20,21,22,23,24,25,26,27,28,29,30,31,32].

Stepwise coordination is a good solution to the first challenge. It involves synthesis of a series of linear and branched π-conjugated bis(terpyridine)metal complex oligomer wires with precise lengths and specific structures on gold or silicon electrodes [33,34,35,36,37,38,39,40]. These wires have shown excellent long-range electron transfer, high redox conductivity, and good redox cyclability, making them promising components for molecular electronics. Elaborate processing of stepwise coordination is not suitable for preparing long wires, as the number of synthetic steps increases proportionally with wire length. Therefore, it is essential to find new, simpler methods to construct long molecular wires.

This work reports the efficient electro-oxidative coupling of aniline-terminated complexes [M(NH_2_-Ph-tpy)_2_] (**1-M**) for preparing π-conjugated M(tpy)_2_ polymer wires with azobenzene bridging (**2-M**) on carbon electrodes (Figure 1). This method efficiently synthesizes well-ordered, long polymer wires of desired lengths and structures with high intrawire redox conduction [33,34,35,36,37,38,39,40]. Moreover, hetero-metal-complex polymer wires ([**2-Fe**]–[**2-Ru**]) applicable to molecular-scale diodes and redox memories can be synthesized readily by sequential electrochemical reactions. Aside from the capacity to produce well-controlled structures and high-performance molecules, the present method is useful for modification of carbon materials through covalent bonding with polymer wires, which is a significant area of current research, given the importance of carbon nanotubes and graphene [41,42,43,44]. Here, we describe the electrochemical synthesis, characterization, and electrochemical properties of π-conjugated homo- and hetero-metal-complex linear polymer wires to demonstrate the general versatility of this method to construct polymer wire arrays with integrated functionalities on carbon.

## 2. Results and Discussion

### 2.1. Electrosynthesis of Polymer Wires

To synthesize **2-Fe** and **2-Ru** on carbon (Figure 1), a MeCN–H_2_O solution of **1-Fe** or **1-Ru** and 0.1 M HClO_4_ was used. Reactions were carried out with consecutive potential scans from −0.5 to 1.4 V vs. Ag/AgCl for **2-Fe** and from −0.3 to 1.6 V for **2-Ru**. The first potential cycle of **1-Fe** showed an irreversible anodic wave at 0.94 V and a reversible wave at 1.09 V (Figure 2A). In contrast, that of **1-Ru** showed three irreversible anodic waves at 0.97, 1.06, and 1.13 V and one reversible wave at 1.24 V (Figure 2B). The irreversible peaks in both cases can be attributed to electro-oxidation of the aniline moiety, and reversible waves resulted from the [M(tpy)_2_]^3+/2+^ couple. Repeated potential scans increased the current signal of the [M(tpy)_2_]^3+/2+^ couple for both **1-Fe** and **1-Ru**, indicating continuous growth of polymer films on the carbon electrode. Peak currents increased almost linearly with the number of scans up to 150 cycles, indicating that film thickness can be readily controlled electrochemically (Figure 2, inset). When the potential sweeps reached 220 cycles, the peak current increase began to slow down (Appendix A). After 400 cycles, the peak currents almost stopped increasing, determining the length of the longest molecular wire that can be synthesized in this way (Appendix A).

After electropolymerization, GC electrodes were covered with a strongly adhered film, which was dark purple in the case of **2-Fe** and red for **2-Ru**. Note that Ru(tpy)_2_ polymer wires are not easily prepared by the stepwise coordination method because the complexation of Ru^2+^ and terpyridine does not proceed at room temperature. HClO_4_ was added to the solution of **1-Fe** or **1-Ru** to protonate the aniline moiety and to shift its oxidation potential positively toward the oxidation potential of M(tpy)_2_. This improves the ability to control electropolymerization. Furthermore, the solubility of protonated complexes is increased in the MeCN–H_2_O solution.

We previously reported the electro-oxidative polymerization of an aniline-attached zinc porphyrin derivative in Bu_4_NClO_4_–CH_2_Cl_2_ to give azobenzene-bridged porphyrin polymer wires on GC and indium tin oxide (ITO) electrodes [45]. The polymer film thickness in the present case (vide infra) was much thicker than that in the previous case, implying that the electrolyte conditions employed in this study and the redox property of the starting materials are more suitable to the formation of ultralong redox polymer wires, whereas their bridging structure is similar to that in the previous case.

### 2.2. Structural Characterization and Electropolymerization Mechanism

**2-Fe** was characterized by Raman spectroscopy and compared with a chemically synthesized linear polymer, [Fe(tpy-AB-tpy)]_n_(BF_4_)_2n_ (**6**), by the coordination reaction of trans-tpy-Ph-N=N-Ph-tpy (**5**) with Fe(BF_4_)_2_·6H_2_O (Figure 3). Characteristic Raman signals of azobenzene [46], Ar–N stretching (1145 cm^−1^), and trans-N=N stretching (1445 cm^−1^) are present in the spectra of both **2-Fe** and [Fe(tpy-AB-tpy)]_n_(BF_4_)_2n_. These two spectra are perfectly consistent with each other, and no extra peaks appear. This result indicates that the electrochemically synthesized polymer has a linear azobenzene-bridged structure.

Here, we propose the most plausible mechanism of the electropolymerization reaction in this study. The literature records two possible ways to form azobenzene by oxidation of aniline. One is the condensation reaction of aniline and nitrosobenzene produced by oxidation of aniline [47]. The other occurs via oxidation of diphenylhydrazines stemming from homocouplings of aniline radicals, which are common in electrochemical reactions [48]. In the present case, the anchoring reaction of molecular wires starts from electro-oxidation of an amino group of the monomers (Step I, left part in Figure 4). In the present electropolymerization, the corresponding oxidation waves appear at 0.94 V (**1-Fe**) and 0.97 V (**1-Ru**) in the cyclic voltammograms [49,50]. These nitrogen radicals react with the carbon electrodes to immobilize themselves through radical addition to a π-bond of the carbon, thus generating a geminal carbon radical (Step II) [51]. This surface-immobilized nitrogen is further oxidized and reacted with the geminal carbon radical to form a N–C–C three-membered ring (Step III). This stable anchor has been identified in a similar reaction on fullerene [52] and is supported by results of DFT calculations (see Appendix A).

Elongation of the wires starts immediately after stable surface bonding is established. The other aniline group of the surface complex is converted into a radical by oxidation (Step IV, Figure 4). The radical seizes one electron from an aniline group of the nearest complex in solution, and further oxidation generates a pair of radicals in close proximity. Due to their high reactivity and closeness, they couple to form an N–N bond before the radical in solution diffuses far away (Step V). This diphenylhydrazine linkage is then oxidized to form azobenzene through a 2e^−^ oxidation reaction (Step VI) [53]. Oxidation waves appear at 1.06 and 1.13 V in the cyclic voltammogram of **1-Ru** (Figure 2B), while these two peaks overlap with the anodic wave of [Fe(tpy)_2_]^3+/2+^ for **1-Fe**. Finally, repeating Steps IV, V, and VI generates long polymer wires.

### 2.3. Electrochemical Performance and Morphology of Electropolymerized Films

Figure 5A shows a cyclic voltammogram of **2-Fe** in Bu_4_NClO_4_–MeCN, in which two reversible waves for [Fe(tpy)_2_]^3+/2+^ and [Fe(tpy)_2_]^2+/1+^ appear at 0.79 and −1.68 V vs. Ag/Ag^+^, respectively [54]. The cyclic voltammogram of **2-Ru** in Bu_4_NClO_4_–MeCN displays three reversible waves for [Ru(tpy)_2_]^3+/2+^, [Ru(tpy)_2_]^2+/1+^, and [Ru(tpy)_2_]^1+/0^ at 0.96, −1.68, and −1.90 V, respectively (Figure 5C) [54]. The peak currents of [Fe(tpy)_2_]^3+/2+^ in **2-Fe** and [Ru(tpy)_2_]^3+/2+^ in **2-Ru** increase linearly with the scan rate (Figure 5B,D). The linear relationship in both cases confirms their surface-confined nature and also indicates that redox reactions in these polymer wires are facile (vide infra).

In the present case, coverage of polymer wires, Γ (mol cm^−2^), was evaluated experimentally as follows (Figure 6). First, a **2-Fe**-modified GC electrode was treated with methanolic KOH solution under an applied potential to oxidize and dissociate Fe^2+^ ions, leaving noncoordinating terpyridine moieties on the electrode. The KOH–MeOH solution prevents the formation of insoluble iron oxides [55]. Electrochemical measurements showed that more than 99.9% of the complexes were removed (Figure 7B). The electrode was then treated with an ethanolic Fe(BF_4_)_2_·6H_2_O solution, followed by a solution of terpyridine in CHCl_3_ to form a monolayer of Fe(tpy)_2_. Finally, the Γ value was evaluated from the cyclic voltammogram of [Fe(tpy)_2_]^3+/2+^. The Γ value thus obtained was 2.00 (± 0.18) × 10^−11^ mol cm^−2^ (Figure 7C). This value is more similar to the case of strong Si–C bonding formed by hydrosilylation on silicon, 3–4 × 10^−11^ mol cm^−2^ [34,36], rather than the case of modest Au–S bonding on gold, 1.0–1.2 × 10^−10^ mol cm^−2^ [39,40]. The maximum coverage of [Fe(tpy)_2_]^2+^ on a surface evaluated by a close-packing model is 1.6 × 10^−10^ mol cm^−2^. 

The lengths of polymer wires were evaluated from their cyclic voltammograms based on the Γ value of the monolayer noted above. The Γ value of Fe(tpy)_2_ of the **2-Fe** sample synthesized by 150 potential scan cycles was 3.3 × 10^−8^ mol cm^−2^, which corresponds to ca. 1500 units and a wire length of 3 μm based on molecular modelling, indicating that the length of one Fe(tpy)_2_ unit is 2 nm (see Appendix A). The surface coverage of Ru(tpy)_2_ in the **2-Ru** sample synthesized by 150 potential scan cycles was 2.2 × 10^−8^ mol cm^−2^, corresponding to ca. 1000 units and a length of 2 μm.

The surface morphology of a GC electrode modified with **2-Fe** (average length: 3 μm) was analyzed with atomic force microscopy (AFM). Its topography had height differences between peaks and troughs of approximately 20 nm, which was only 0.6% of the wire length (Figure 8B), indicating that all wires have an identical structure and completely cover the electrode surface. On the other hand, the phase image of **2-Fe** shows a uniform pattern with a domain size of around 60 × 30 nm (Figure 8D), which is common in wire-type materials and different from that of pristine GC (Figure 8C). 

### 2.4. Charge Transport in Long Fe(tpy)_2_ Polymer Wires

The highest coverage of Fe(tpy)_2_ units in **2-Fe** obtained by the electropolymerization method was 1.63 × 10^−7^ mol cm^−2^ (Figure 10A). This corresponds to 7410 layers and an average molecular wire length of 14.8 μm. The surface structure and length of this longest **2-Fe** were observed with optical microscopy (Figure 9). The wire length, as measured from a side-view image of the electrode, was around 15 μm, which is almost identical to the value determined electrochemically. However, wires protruding from the GC electrode formed a rough surface, which was different from the smooth surface of the 3 μm sample measured with AFM (above). We expect that the self-supporting force of individual wires weakens, resulting in the formation of large bundles and a rough surface when the wires grow longer. 

An important property of molecular wires is their charge transport capacity [10]. Here, we employed chronoamperometry and chronocoulometry to evaluate the charge transport kinetics of **2-Fe** and **2-Ru** in 1 M Bu_4_NClO_4_–MeCN. The chronoamperogram of the longest polymer wires (close to 15 μm) shows that both oxidation and reduction currents decreased to zero within 5 s (Figure 10B). The charge transport diffusion coefficient (*D*_ct_) can be derived from the linear region between 0.85 s^−1/2^ (1.4 s) and 3.54 s^−1/2^ (0.08 s) in the plot of *i* vs. *t*^−1/2^ (Figure 10C). The *D*_ct_ value is 4.6 ×10^−7^ cm^2^ s^−1^, which is remarkably higher than that of other redox complex polymer films: 2.2 × 10^−10^ to 4.5 × 10^−9^ cm^2^ s^−1^ for poly-[Ru(vbpy)_3_]^2+^ (vbpy: 4-vinyl-4’-methyl-2,2’-bipyridine) [56], 1.6 × 10^−8^ cm^2^ s^−1^ for a copolymer of [Os(vpy)(bpy)_2_Cl]^+^ (vpy: 4-vinylpyridine and bpy: 2,2’-bipyridine) [57], 1 × 10^−7^ to 1 × 10^−8^ cm^2^ s^−1^ for PVP–Fe(CN)_5_ (PVP: poly(4-vinylpyridine)) [58], and 1 × 10^−8^ to 1 × 10^−9^ cm^2^ s^−1^ for polyvinylferrocene [59,60]. A possible reason for this high redox conductivity of **2-Fe** is that these molecular wires have highly π-conjugated structures, which give them facile through-bond redox conduction between the Fe(tpy)_2_ moieties. We previously reported this behavior for oligomeric wires prepared by the stepwise coordination method [5], but this is our first time seeing such a high value in micrometer-length wires. Note that the current-decay profile of this polymer wire differs from that of Fe(tpy)_2_ oligomer wires on a gold surface, which shows a rate-determining step of current flow arising from charge transportation between the electrode surface and the first-layer complex via Au–S bonds, which cause the current flow not to obey the Cottrell equation [39]. In the present case, the anchor of the polymer wire on carbon via a N–C–C three-membered ring with a unique conjugate structure enables fast electron transfer between the electrode and the nearest Fe(tpy)_2_ unit.

### 2.5. Hetero-Metal-Complex Wires 

Hetero-metal-complex polymer wires attract much attention owing to their unique electronic properties and potential applicability as, for example, single-molecule diodes and single-molecule memory [61]. We employed the present electropolymerization method to construct hetero-metal-complex polymer wires, GC/[**2-Fe**]–[**2-Ru**] (**3**) and GC/[**2-Ru**]–[**2-Fe**] (**4**). The cyclic voltammograms of the first preparation step of the hetero wires appear no different from those of homo wires. However, the anodic and cathodic peak currents of the second block show unequal current increases as the second block grows longer. **3** shows a clear cathodic peak, but a small anodic peak derived from Ru-wires in the second step voltammogram, and the cathodic peak grows visibly as the scan number increases (Figure 11A). On the other hand, **4** shows a sharp anodic peak and a broad cathodic peak in the potential region of Fe-wire redox reaction in the second-step voltammogram, and the anodic peak grows faster than the cathodic one (Figure 11B). In both cases, redox responses of the first block show almost no decrease during the growth of the second block on top of the first. The unequal increasing rates of the anodic and cathodic peak currents indicate that the second complex wire is connected to the top of the first wire, and not directly to the carbon electrode. 

The cyclic voltammograms of **3** in Bu_4_NClO_4_–MeCN at a scan rate lower than 10 mV s^−1^ show two pairs of redox waves at 0.79 and 0.96 V vs. Ag/Ag^+^ attributable to [Ru(tpy)_2_]^3+/2^ and [Fe(tpy)_2_]^3+/2^, respectively (Figure 11C). The peak currents of cathodic and anodic waves of **2-Fe**, in the first block of the wire, are close to each other, whereas the cathodic peak of **2-Ru**, in the second block, is sharper and higher than the anodic one. When the scan rate is increased to 100 mV s^−1^, the cathodic peaks of both blocks remain sharp and gradually fuse. On the other hand, the sharpness of the anodic peaks of **2-Fe** shows almost no change, but that of **2-Ru** becomes much broader and finally inseparable above 100 mV s^−1^.

The cyclic voltammograms of **4** show an irreversible phenomenon similar to **3**, but they reflect the reverse connection of the two complex wires (Figure 11D). The first Ru(tpy)_2_ block connected to the GC electrode shows no difference in the cathodic and anodic waves, while the second Fe(tpy)_2_ block shows considerable dependence of the anodic peak current and sharpness on the scan rate. The unequal cathodic and anodic waves in each hetero wire are “redox diode” behaviors that can be attributed to a redox potential difference of 0.17 V between [Ru(tpy)_2_]^3+/2+^ and [Fe(tpy)_2_]^3+/2+^ (Figure 5). Redox diode behavior is observed in hetero redox polymer films [62,63]. In the cyclic voltammogram of **3**, the [Fe(tpy)_2_]^3+/2+^ couple attached directly to the electrode has similar oxidation and reduction rates, whereas the [Ru(tpy)_2_]^3+/2+^ couple shows slow oxidation and fast reduction because the electron transfer is perturbed by the inner [Fe(tpy)_2_]^3+/2+^ couple. In the wire with the reverse structure, **4**, the inner Ru(tpy)_2_]^3+/2+^ couple controls the electron transfer of the outer [Fe(tpy)_2_]^3+/2+^ couple, resulting in slow reduction and fast oxidation. Note that similar π-conjugated hetero-metal oligomer wires prepared by the stepwise coordination method did not show such diodelike behavior clearly [35,40]. This implies that the length of the molecular wires in this study is crucial for a significant hetero-junction effect on the direction of electron transfer.

## 3. Conclusions

A highly efficient method for preparing long polymer wires on a carbon surface was established by using electrochemical coupling reactions of anilinoterpyridine complexes of iron and ruthenium. The resulting wire consists of a π-conjugated linear and rigid structure with azobenzene bridging. This electropolymerization method affords ultralong (up to 7400 units corresponding to 15 μm) bis(terpyridine)iron and bis(terpyridine)ruthenium polymer wires, **2-Fe** and **2-Ru**, respectively, with fast redox conduction capacities. Hetero-metal-complex polymer wires on a glassy carbon electrode, GC/[**2-Fe**]–[**2-Ru**] and GC/[**2-Ru**]–[**2-Fe**], were prepared by two-step sequential electropolymerization reactions. The cyclic voltammograms of these polymer wires show redox diodelike behaviors due to the redox potential difference at the junction of two blocks. Our method offers an easy way to synthesize polymer wire arrays and to integrate functional molecules on carbon.

## Figures and Tables

**Figure 1 molecules-26-04267-f001:**
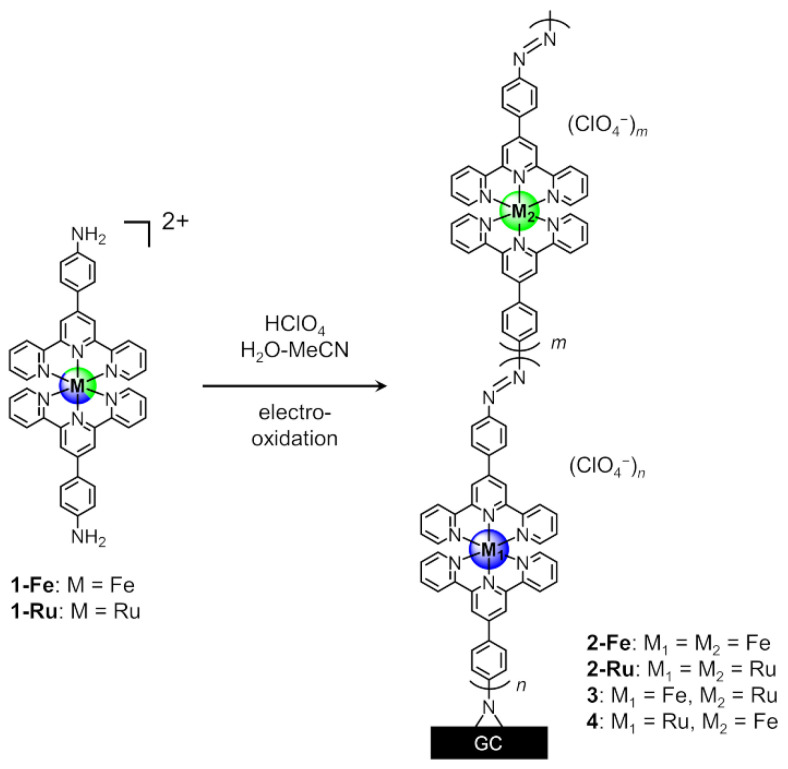
Electrosynthesis of azobenzene-bridged M(tpy)_2_ polymer wires.

**Figure 2 molecules-26-04267-f002:**
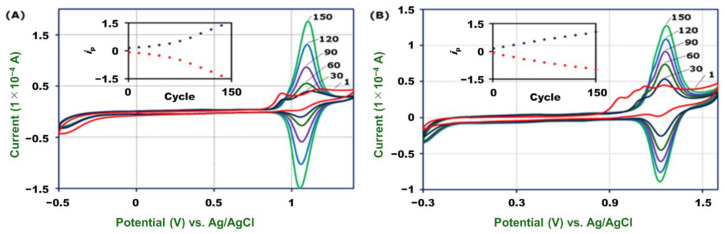
Cyclic voltammograms for electropolymerization of 1 mM **1-Fe** (**A**) and **1-Ru** (**B**) on GC (3 mm ø) in HClO_4_–MeCN–H_2_O at a scan rate of 100 mV s^−1^. The number of cyclic potential scans is labeled. Inset: plots of peak current (*i*_p_) vs. the number of potential scans.

**Figure 3 molecules-26-04267-f003:**
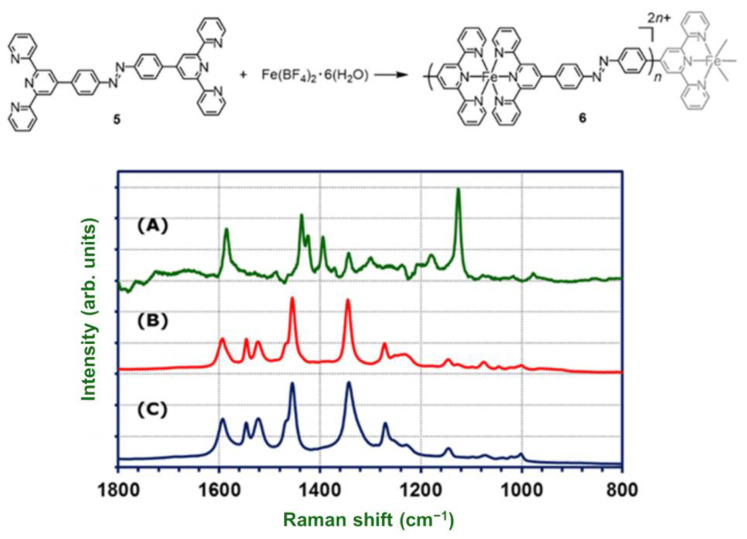
Raman spectra of trans-tpy–Ph–N=N–Ph–tpy (**5**) (**A**); chemically synthesized polymer, [(tpy-AB–tpy)Fe]_n_ (BF_4_)_2n_ (**6**) (**B**); and electrochemically synthesized polymer, **2-Fe** (**C**).

**Figure 4 molecules-26-04267-f004:**
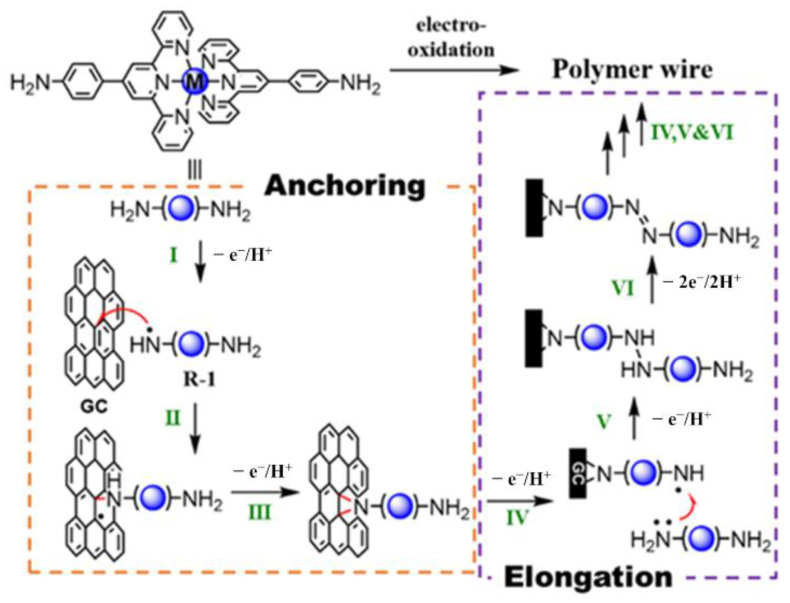
Possible mechanism of the electrosynthesis of M(tpy)_2_ polymer wires.

**Figure 5 molecules-26-04267-f005:**
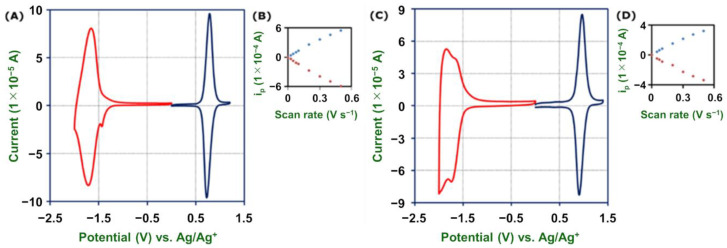
(**A**) Cyclic voltammograms of **2-Fe** on GC in 1 M Bu_4_NClO_4_–MeCN at 100 mV s^−1^. (**B**) Peak current of the redox wave of [Fe(tpy)_2_]^3+/2+^ vs. the scan rate. (**C**) Cyclic voltammograms of **2-Ru** on GC in 1 M Bu_4_NClO_4_–MeCN at 100 mV s^−1^. (**D**) Peak current of the redox wave of [Ru(tpy)_2_]^3+/2+^ vs. the scan rate.

**Figure 6 molecules-26-04267-f006:**
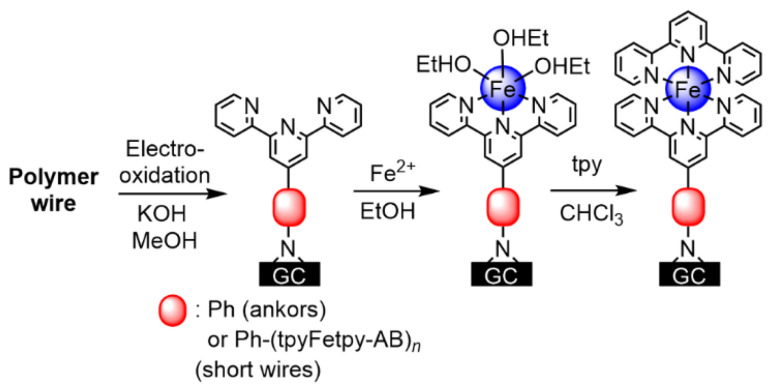
Electrochemical dissociation of Fe(tpy)_2_ polymer wires, followed by Fe(tpy)_2_ monomer formation.

**Figure 7 molecules-26-04267-f007:**
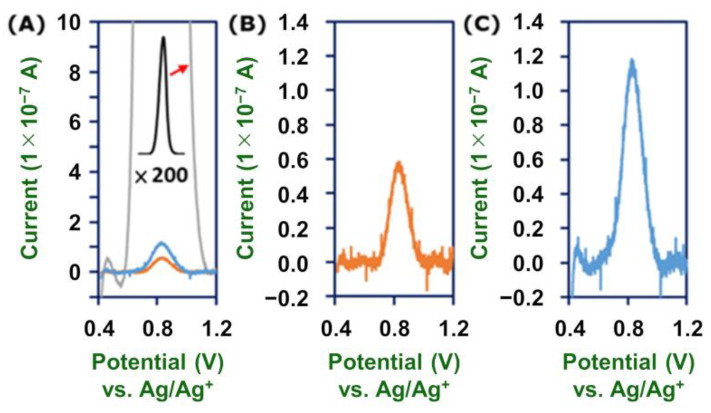
(**A**) Linear sweep voltammograms of an electrode with **2-Fe** (gray) after oxidative treatment with methanolic KOH solution (orange) and after Fe(BF_4_)_2_·6H_2_O and terpyridine treatment (blue). (**B**,**C**) Isolated second and third voltammograms from (**A**).

**Figure 8 molecules-26-04267-f008:**
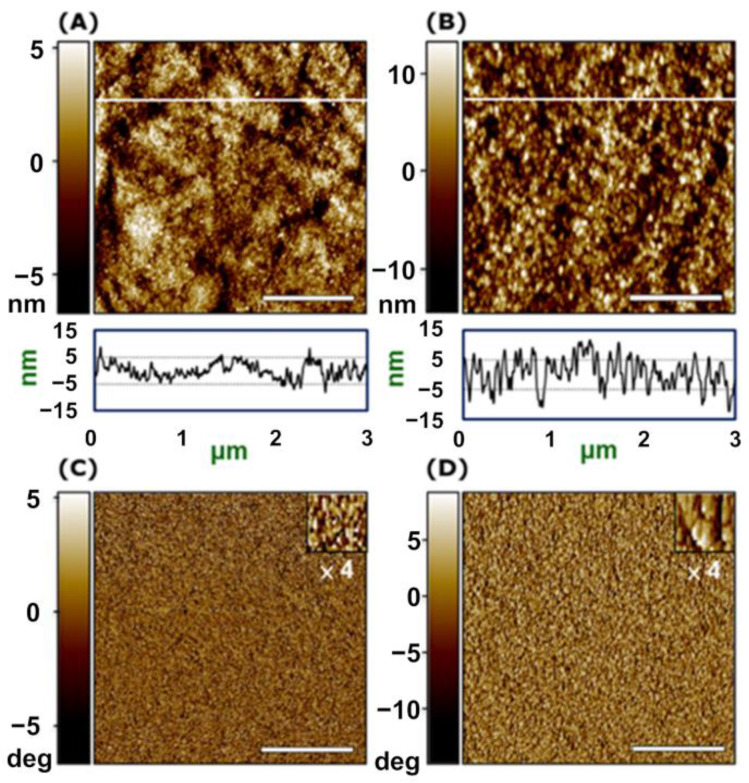
AFM topography and phase images: (**A**,**C**) pristine GC and (**B**,**D**) **2-Fe**-modified electrode. Scale bars are 1 μm.

**Figure 9 molecules-26-04267-f009:**
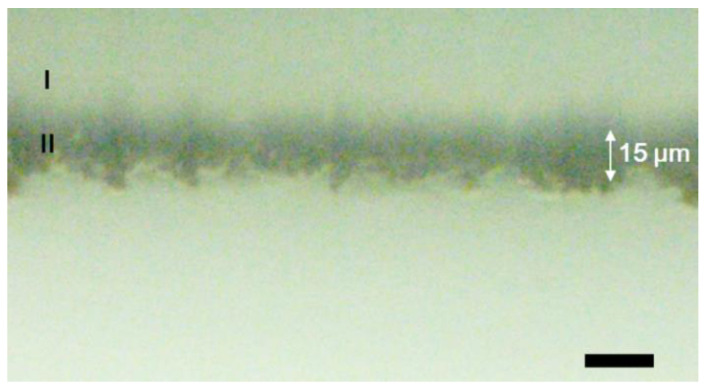
Side-view optical microscopy image of an electrode (region I) modified by the longest Fe(tpy)_2_ polymer wires (region II). Scale bar is 20 μm.

**Figure 10 molecules-26-04267-f010:**
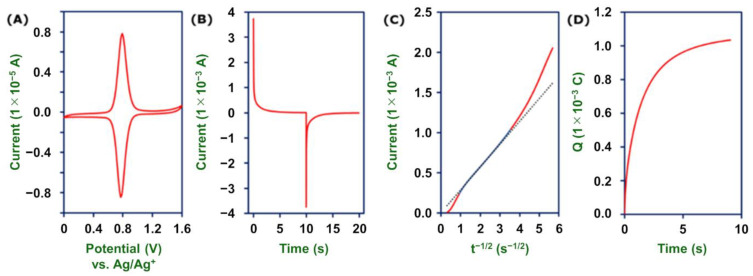
Electrochemical measurements of the longest **2-Fe** wires (Γ = 1.63 × 10^−7^ mol cm^−2^) on GC in 1 M Bu_4_NClO_4_–MeCN. (**A**) Cyclic voltammogram at a scan rate of 1 mV s^−1^. (B) Chronoamperogram with a potential step from 0.5 to 1.1 V. (**C**) *i*-*t*^−1/2^ plot derived from (**B**). The dotted line is the linear fit of the blue region, *R*^2^ = 0.9954, *y* = 2.854 × 10^−4^
*x*. (**D**) Chronocoulogram with a potential step from 0.5 to 1.1 V.

**Figure 11 molecules-26-04267-f011:**
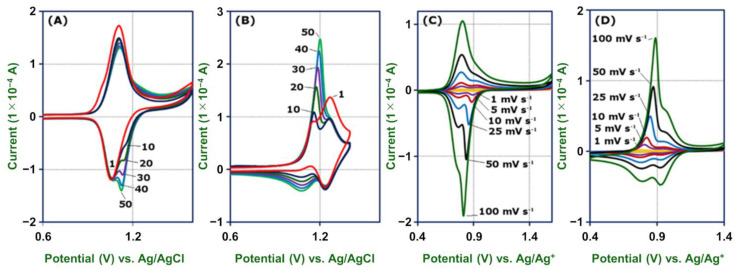
Cyclic voltammograms of the electrosynthesis of **3** (**A**) and **4** (**B**). Cyclic scans are labeled in order. Cyclic voltammograms of **3** (**C**) and **4** (**D**) in 1 M Bu_4_NClO_4_–MeCN at various scan rates as shown in the figure.

## Data Availability

Not applicable.

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
