# Peer review of "Ultralong π-Conjugated Bis(terpyridine)metal Polymer Wires Covalently Bound to a Carbon Electrode: Fast Redox Conduction and Redox Diode Characteristics"

_molecules, 2021, doi:10.3390/molecules26144267_

Round 1

Reviewer 1 Report

In the paper the authors developed an efficient and convenient electrochemical method to synthesize π-conjugated redox metal-complex linear polymer wires composed of azobenzene-bridged bis(terpyridine)metal (2-M, M = Fe, Ru) units covalently immobilized on glassy carbon (GC). This work provides a new way of synthesize  π-conjugated redox metal-complex linear polymer wires which could attract interests of chemists especially who work on polymers. I think the paper is qualified to be published on this journal, with only a few remarks for the authors:

  1. The authors provide the optical microscopy image of the wire, why not provide some SEM images?
  2. Is it possible to provide some discussion on the length limit of the synthesized polymer wire?

Author Response

Reply to Reviewer 1:

We thank Reviewer 1 very much for the comments, which give us an opportunity to improve our manuscript.

Comment 1:

The authors provide the optical microscopy image of the wire, why not provide some SEM images?

Answer:

Regretfully, we have not measured SEM image of the electropolymerized films on GC, and it is difficult to measure the images at present as all the samples are consumed. We think that the optical microscope images and AFM images are sufficient to analyze the thickness and morphology of the films.

Comment 2:

Is it possible to provide some discussion on the length limit of the synthesized polymer wire?

Answer:

We added the following two figures in Supplementary Materials.

Figure S1. Cyclic voltammetry of 1-Fe (1 mM) at a GC electrode in MeCN-H2O-HClO4 at a scan rate of 0.1 V s-1. Numbers in the figure refer to those of the cyclic scans.

Figure S2. Relation of the oxidation and reduction peak currents in the cyclic voltammograms for electropolymerization of 1-Fe (1 mM) at a GC electrode in MeCN-H2O-HClO4 at a scan rate of 0.1 V s-1 versus the number of the cyclic scans.

Based on the results in these figures, we added the discussion on the length limit of the polymer wire on page 2, last 4 lines as follows.

“When the potential sweeps reached 220 cycles, the peak current increase begins to slow down (Figure S2). After 400 cycles, the peak currents almost stop increasing, determining the length of the longest molecular wire that can be synthesized in this way (Figure S2).”

Reviewer 2 Report

A highly efficient method for preparing long polymer wires on a carbon surface was established by using electrochemical coupling reactions of anilinoterpyridine complexes of iron and ruthenium. This electropolymerization method affords ultra-long (up to 7400 units corresponding to 15 μm) bis(terpyridine)iron and bis(terpyr- idine) ruthenium polymer wires. In order to achieve these results, the authors conducted a number of studies on how to obtain log polimer wires. Such wires are the most important components of molecular-scale electronics for molecular or quan- tum computing, therefore the work is very useful for practical use. The data presented are innovative and make significant progress in the field of molecules science. New techniques and technologies were used in the research. Correct conclusions were made based on the results obtained. I rate the work very highly and propose to publish the article in its current form.

Author Response

We thank Reviewer 2 very much for the comments.